# Biomass Novel Adsorbents for Phenol and Mercury Removal

**DOI:** 10.3390/molecules27217345

**Published:** 2022-10-28

**Authors:** Joao Manuel Valente Nabais, Carlos Eduardo Laguinhas, Silvia Román

**Affiliations:** 1Comprehensive Health Research Center (CHRC), 1150-082 Lisbon, Portugal; 2Departamento de Química, Universidade de Évora, Escola de Ciências e Tecnologia, Rua Romão Ramalho n° 59, 7000-671 Évora, Portugal; 3Departamento de Física Aplicada, Universidad de Extremadura, Avda Elvas s/n, 06071 Badajoz, Spain

**Keywords:** activated carbons, adsorption, mercury, phenol, water treatment, biomass

## Abstract

This paper reports the use of activated carbons made from novel agriculture and industrial wastes, namely sunflower, vine shoots, and coffee endocarp, to remove two high-priority contaminants: phenol and mercury species (under different forms) from aqueous solutions. The activated carbons were used as prepared and also modified with nitric acid and triethylenediamine in order to explore additional adsorption mechanisms. The results showed an interesting potential of the materials to be used for water decontamination as indicated by the mercury uptake up to 1104 mg/g for Hg^2+^, 771 mg/g for [HgCl_4_]^2−^, 966 mg/g for HgCl_2_ and the maximum phenol adsorption capacity of 190 mg/g. The modification with triethylenediamine led to a significant increase in the phenol and mercury adsorption reaching an increment of 85% for phenol and 250% for Hg^2+^.

## 1. Introduction

Phenol and mercury are two important toxic materials listed as priority pollutants by the US Environmental Protection Agency (EPA) and also by the European Union [1]. Regarding inorganic mercury, the EPA indicates the erosion of natural deposits, discharge from refineries and factories, and runoff from landfills and croplands as the main sources and limits the highest level in drinking water to 2 μg/L, while in the case of the European Union, this level drops to 1 μg/L [2]. For pentachlorophenol (PCP) this limit is 1 μg/L according to the EPA, which specifies the discharge from wood-preserving factories as the main contaminant source (the EU sets the limit at 5 μg/L [2]). The World Health Organisation (WHO) guidelines for drinking water quality recommend a limit of 6 μg/L for inorganic mercury and 9 μg/L for pentachlorophenol [3].

Phenol is considered to be very toxic to humans through oral exposure with symptoms including muscle weakness and tremors, loss of coordination, paralysis, convulsions, coma, liver and kidney damage, headache, fainting, and other mental disturbances. The ingestion of 1 g has been reported to be lethal. Inhalation and dermal exposure to phenol are highly irritating to the skin, eyes, and mucous membranes in humans. Mercury and its compounds are present both in river and ocean water and their presence is associated with atmospheric deposition [4]. These compounds act as dangerous and insidious poisons, with the possibility of adsorption not only through the gastrointestinal tract but also through the skin and lungs; and Hg accounts for 250,000 intellectual disabilities annually. After adsorption, mercury circulates in the blood and is stored in the liver, kidneys, brain, spleen, and bones, thereby leading to several health problems such as paralysis, serious intestinal and urinary complications, dysfunction of the central nervous system and in more severe cases of intoxication, death. The soluble compounds of mercury are particularly toxic because their adsorption is very fast; ingesting a dose of less than 0.5 g can prove to be fatal.

The reduction of phenol and mercury discharges and the removal of pollutants from water streams are still needed, despite the number of papers already published on this topic. The more stringent regulations have led to extensive research on this topic that continues to be carried out. Amongst the methods available for the removal of heavy metals or phenolic compounds in liquid solutions, the use of activated carbons (ACs) is worth mentioning as it can be considered a cost-effective method due to the high removal efficiency and the flexibility of the industrial unit operation [5,6,7,8,9,10]. The search for ACs precursors that are abundant and with low prices is still a matter of research, now fostered due to the political strategies aimed to promote the Circular Economy.

The uptake of adsorbate from aqueous solutions by ACs is complex; it depends on various factors, which include the type of precursor, the physical nature (surface area, pore size, pore volume, ash content, particle size) and functional groups present on the adsorbent, the nature of adsorbate (pKa, polarity, molecular weight, size, solubility) as well as the solution conditions (pH, temperature) [10,11]. This makes it necessary to perform specific analysis on any specific adsorbent; the adsorption mechanisms can only be suggested once the ACs have been tested under varying experimental conditions.

In this context, this work aimed to evaluate the adsorption of phenol and mercury from dilute aqueous solutions onto ACs produced from agriculture and industrial wastes, namely vine shoots, sunflower, and coffee endocarp, and find the interactions governing the adsorption on each particular system. These three precursors meet the requirements of being abundant in the southwest of the Iberian Peninsula and are biomass wastes that have been scarcely investigated to produce ACs. The thoughtful textural and chemical characterization of the adsorbents, and their subsequent impregnation with triethylenediamine, to further explore the enhancement of adsorption by the participation of additional surface groups, was related to removal capacities and optimal preparing conditions were identified.

## 2. Results and Discussion

### 2.1. Characterization of Materials

Table 1 lists typical porosity parameters obtained by applications of classical models (BET, Dubinin-Radushkevitch, and alfa model, as described in Section 3.2) to N_2_ adsorption experimental data. The surface net acidity, as estimated from the point of zero charge, has also been included in this Table. All pristine samples (i.e., without pretreatment) had basic properties with a point of zero charge (pH_pzc_) between nine and ten, as can be seen in Table 1. It is noteworthy to highlight that the modification with TEDA did not change significantly the pH_pzc_ value. On the contrary, the oxidation with nitric acid, as expected, caused a significant decrease in the basicity of the carbons with the pH_pzc_ changing from 9.71 to 2.32.

The surface chemistry characterisation conducted by FTIR and representative spectra are shown in Figure 1, indicating that the ACs produced by carbon dioxide and water vapour activation have similar surface functional groups as shown by the presence of comparable absorption bands in the spectra.

The water vapour activation samples show less intense bands which corresponds to a smaller concentration of surface groups. In addition to the characteristic bands of the aromatic structure, the FTIR spectra show the presence of quinones, lactones, carbonyls, phenols, alcohols, and pyrones as indicated by the existence of the bands at 1700–1900 cm^−1^ (ν(C = O)) and 1450–1420 cm^−1^ (ν(C-O)). The bands attributed to alcohol and phenol are situated around 3500 cm^−1^, a region not shown here for simplicity. The modification of the ACs with TEDA leads to a higher presence of electron-donating groups such as pyrones, as indicated by the increased intensity of the correspondent bands, and the occurrence of new functional groups from TEDA, such as amines. On the contrary, bands at 990–1120 cm^−1^, attributed to ν(C-O) in hydroxyl groups or ether-type structures are less intense. The oxidation with nitric acid has introduced new acidic groups, coherent with the pH_pzc_ decrease, such as carboxylic acid and anhydride as indicated by the presence of bands 1651 and 1697 cm^−1^ and 1740–1880 cm^−1^, respectively. A more detailed analysis and the spectra can be found elsewhere [12].

The nitrogen adsorption/desorption isotherms for all samples can be classified as type I according to IUPAC [13], not shown here for simplicity, which indicates ACs with a predominant porous structure composed of micropores. As shown in Table 1, where typical porosity parameters have been listed, all samples have low external area reaching a maximum of 71 m^2^g^−1^ for sample G742. Regarding the pristine activated carbon samples, we can see a reasonable porous development with BET apparent surface areas between 308 and 956 m^2^g^−1^ and total micropore volume from 0.15 to 0.44 cm^3^g^−1^. The values of burn-off attained were in all cases less than 50%wt, which is suitable for industrial production of ACs. Additionally, for a particular precursor, the increase in the temperature produces samples with a higher burn-off value and an enhanced porosity development (with the exception of G819). The modification with TEDA has led to a general increase of the external area and a decrease of the BET apparent surface area within the range of 39 to 165 m^2^g^−1^. The same trend can be observed for the micropore volume. The decrease in porosity can be associated with the formation of clusters that partially block the micropores [14,15,16]. The oxidation with nitric acid has caused a greater impact on the sample’s porosity, namely on the BET apparent surface area and micropore volume with a decrease of 310 m^2^g^−1^ and 0.15 cm^3^g^−1^, respectively. The similarity between Vs and V_0_ indicates that samples, in general, have pores in the range of primary micropores with a mean pore width of less than 0.7 nm.

All activated carbon samples are in powder format. The microstructural analysis was conducted by SEM, representative micrographs shown in Figure 2, show that AC samples mainly retain the distinct structure of the precursors used.

It is interesting to observe the differences between precursors, as revealed in Figure 2, with vine shoots AC with a large canal that gives access to smaller ones. Sample G77W shows a much more regular structure when compared with others. On the other hand, the coffee endocarp sample shows a structure similar to a stack of tubes.

### 2.2. Phenol Adsorption

The phenol adsorption isotherms onto the AC samples can be classified as type L according to the Giles classification, suggesting that the adsorption occurs mainly via the pi-pi interactions between the aromatic structure of the carbon and the aromatic ring of the phenol molecules, which most probably are aligned parallel to the carbon surface [17]. This hypothesis is strengthened by the fact that at pH 6, the predominant phenol species in solution is the molecular form [10]. Therefore, the dispersive interactions are expected to be a relevant driving force for the adsorption of phenol along with other possible factors, such as the specific chemical interactions or the porous structure of the carbon materials as in most cases the adsorption is a complex process that needs to be analysed case by case.

The maximum adsorption capacity, taken from the adsorption isotherms, ranges from 47 mg/g to 190 mg/g for samples G819 and G733WT, respectively, as shown in Figure 3. The results obtained are comparable favourably with other published results [18,19,20]; however, this comparison should be made with caution because of the different experimental conditions used that might have an influence on the phenol adsorption.

According to the bibliography and the authors’ own experience, one factor that clearly influences the adsorption performance is the pore volume and pore size distribution of the adsorbents. While the volume has to be enough to provide the adsorbate room to be adsorbed on several layers, if possible, the pore size distribution also has to be suitable to let the molecule access the inner of the carbon matrix and impede blockage, in order to maximize removal efficiency. This effect has been clearly found from our runs, as can be suggested from Figure 4, where both the phenol adsorption capacity and porosity parameters (BET apparent surface, m^2^/g and total pore volume, cm^3^/g) have been plotted as bar height, with a different colour for each run. In addition, in our case, all adsorbents are microporous, as deduced from the low contribution of the external surface (A_EXT_, m^2^/g Table 1) which has been associated with favourable behaviour towards phenol compounds [21]. Phenol adsorption is proportional to the BET apparent surface area and pore volume. All samples have similar surface chemistry and pH_pzc_ but dissimilar adsorption capacity, which indicates that surface chemistry does not have a significant impact on the adsorption. The process is governed by physical adsorption (physisorption) rather than chemisorption.

On the other hand, it is clearly visible that the modification with TEDA lead to a significantly increased adsorption, despite the decrease in the porous structure. Sample V823WT reached a maximum adsorption capacity of 119 mg/g, which constitutes a 65% increase when compared with sample V823W. Regarding the pair G733WT-G733W, the increase was 85%. The BET apparent surface area decreased from 523 to 358 m^2^/g, for the first case, and from 594 to 555 m^2^/g for the sunflower samples.

This fact can have a twofold explanation as follows. On one side, the existence of chemical interactions between phenol and TEDA molecules anchored in the carbon surface mainly via the protonation of the nitrogen groups. On the other side, the modification with TEDA has incorporated into the carbon structure more electron-donor groups, as indicated by FTIR, which in turn enhances the dispersive interaction between phenol and the aromatic structure of the AC samples, leading to an adsorption increase. Figure 5 shows a schematic representation conducted using molecular dynamics methods to illustrate the phenol adsorption onto the ACs before and after the modification with TEDA.

### 2.3. Mercury Adsorption

The study of three different mercury species, namely HgCl_2_, Hg^2+^, and [HgCl_4_]^2−^, is relevant to mimic what happens in the water streams where mercury usually exists in the form of complexed species, which can be negative, neutral, or positive, and is not usually tackled in the bibliography. The control of the mercury species present in each case is also important to understand the adsorption mechanisms of ACs along with other factors, such as the net surface charge of the materials determined by the ionization of the surface functional groups that depend on the ACs’ pH_pzc_ and the pH of the solution. The carbon material acquires a net positive or negative charge when the solution pH is smaller or bigger than the pH_pzc_ value. The pH of each solution is five, three, and seven for the species HgCl_2_, Hg^2+^, and [HgCl_4_]^2−^, respectively, which lead to the ACs’ net surface charge shown in Table 2.

The maximum adsorption capacity, extracted from the adsorption isotherms, for the adsorption of each mercury species is also shown in Table 2.

The adsorption capacity of the pristine ACs is higher for Hg^2+^, probably because of the formation of complexes with surface functional groups such as hydroxyl, phenol, and carbonyl [22,23]. The lower adsorption of [HgCl_4_]^2−^ can be due to its larger molecular size which can complicate the adsorption. If we compare the adsorption of each mercury species onto the pristine ACs it is interesting to see that the porous structure does not fulfil a major role in the adsorption process, apart from making more difficult the adsorption of [HgCl_4_]^2−^, as there is no proportionality between the adsorption capacity and the BET area or the pore volume. The relative impact of the electrostatic interactions between the carbon surface and the charged mercury complexes is very small or inexistent. As can be seen in Table 2, all ACs have positive charged surfaces, under the experimental conditions used, which would be favourable for the adsorption of the negative charged [HgCl_4_]^2−^ and detrimental for Hg^2+^ because of the electrostatic repulsions. However, the adsorption capacity is opposite to this expected behaviour, which indicates that the factors already mentioned have a bigger impact on the adsorption mechanism than the electrostatic interactions.

The oxidation of V840 has led to a significant change in the surface chemistry with the introduction of oxygen acidic groups and a decrease of the pH_pzc_ value to 2.32. This impacts the net surface charge because the pH_pzc_ is for sample V840ox smaller than the working pH of the experiments. The highest value observed for the adsorption of mercury was achieved for Hg^2+^ onto sample V840ox, probably due to the electrostatic attraction between the surface and the mercury specie. On the contrary, the adsorption of [HgCl_4_]^2−^ onto V840ox showed a decrease, when compared with V840, due to the electrostatic repulsion. The oxidation has produced a slight decrease in the porosity and also an increased localization of the aromatic ring electrons by the introduction of more oxygen groups making the structure less prone to interact with HgCl_2_ species via van der Waals interactions, which explains the diminution of the adsorption capacity of V840ox.

Figure 6 shows representative adsorption isotherms for [HgCl_4_]^2−^ and Hg^2+^, such as for the phenol adsorption all isotherms can be classified as type L, according to the Giles classification.

The modification of ACs with TEDA has led to a significant increase in the adsorption capacity of the mercury species, ranging from an increment of 30% for the adsorption of HgCl_2_ onto Cf825T to an increment of 250% for the adsorption of Hg^2+^ onto Cf850T. This fact can be related to the increment of electron-donor groups on the carbon surface, introduced by the impregnation with TEDA, which can be favourable for the adsorption of the mercury species.

The adsorption results found here are promising and further studies might be devoted to investigating the regeneration efficiency associated with each couple adsorbent-adsorbate. In this line, various regeneration methods including gasification, pyrolysis, wet oxygen regeneration, or simple warn desorption could be explored.

## 3. Materials and Methods

### 3.1. Preparation of ACs

The precursors used to produce ACs were vine shoots (V), sunflower (G), and coffee endocarp (Cf). The production process started with the carbonization, carried out at 400 °C for 1 h under a constant N_2_ flow of 85 cm^3^min^−1^, followed by the activation step conducted with carbon dioxide activation at 700 °C and 800 °C under a constant CO_2_ flow of 85 cm^3^min^−1^. The activation was also carried out with water vapour at 700 and 800 °C under a constant N_2_ flow of 85 cm^3^min^−1^ saturated with water vapour; this was accomplished by passing the N_2_ flow by a water reservoir that was heated to its boiling point with a flow rate of 0.2 g min^−1^.

A heating rate of 10 °Cmin^−1^, a horizontal tubular furnace was used for the total procedure. At the end of the activation step, the samples were allowed to cool down below 50 °C under N_2_ flow. Afterward, the ACs were washed with 1000 mL of distilled water for 24 h with stirring at room temperature and then were oven-dried at 110 °C for another 24 h and stored in sealed flasks. Each sample was identified with a code name indicating the type of precursor (M), the temperature of activation (T—7 or 8 for 700 or 800 °C, respectively), and burn-off (BO). The BO was determined as the relation between the mass consumed in reference to the initial char mass; char solid yield for the carbonization conditions set here was around 50%.

For example, sample V870 means a sample with 70% burn-off produced from vine shoot by activation with carbon dioxide at 800 °C. The samples produced by water vapour activation were named by including the letter W at the end of the code, for example, V823W. Sample V840 was oxidised in the liquid phase with concentrated nitric acid; the oxidized sample is designated V840ox, and details can be found elsewhere [12]. Samples Cf825, Cf850, V823W, and G733 were impregnated with triethylenediamine (TEDA, 5%wt.) using the sublimation process described in US Patent 5792720 [24] and named Cf825T, Cf850T, V823WT, and G733T, respectively.

### 3.2. Liquid Phase Adsorption

Batch adsorption experiments were carried out in a series of Erlenmeyer flasks at 25 °C in a shaking thermostat for 240 min using 0.05 g of ACs and 25 mL of phenol or mercury solutions with variable concentration. For the phenol adsorption, solutions with concentrations 0.1 to 2 mmol/L were prepared from a stock solution of 1 × 10^−2^ mol/L (>99%, Aldrich), pH of all solutions 6. The adsorption of three mercury species were studied, namely [HgCl_4_]^2–^, HgCl_2_ and Hg^2+^, using the concentration range of 10–1500 mg/L prepared from a 1500 mg/L stock solution of HgCl_2_ (p.a. grade from Riedel-de-Haën) or Hg(NO_3_)_2_.H_2_O for Hg^2+^ (p.a. grade from Ridel-de-Haën). The control of the mercury species present in solution was accomplished by controlling the chlorine concentration as follows: pCl = 0 for [HgCl_4_]^2–^, pCl = 4 for HgCl_2_, and pCl = 7 for Hg^2+^, and details are given elsewhere [23,25]. The pH of the solutions was not adjusted and was three, five, and seven for the aqueous solutions of the Hg^2+^, HgCl_2_, and [HgCl_4_]^2−^, respectively. The quantification of the phenol and mercury was conducted by UV-Vis spectroscopy carried out on a Thermo UV-Vis spectrophotometer at 270 and 230 nm, respectively [10,22]. The procedure for the mercury determination involved adjusting the pCl to zero to have all mercury in the complex structure, measurements details have been given previously [23].

### 3.3. Characterization of the Materials

Nitrogen adsorption isotherms at 77 K were determined using a Quantachrome Instruments Quadrasorb *SI* after outgassing the samples at 400 °C in a Quantachrome MasterPrep Unit. From N_2_ adsorption data, the analysis of the isotherms by the Brunauer-Emmett-Teller (BET), Dubinin-Radushkevich (DR)s and alfa-s methods, allows us to determine the BET apparent surface area (A_BET_), the total micropore volume (V_s_), the external area (A_ext_), and the primary micropore volume (V_0_).

The point of zero charges was determined by mass titrations using a suspension with 7% (*w/v*) in carbon material in NaNO_3_ 0.1 M as described in previous works [26]. Scanning electron microscopy (SEM) was carried out using a JEOL, model JSM-6300, microscope. FTIR spectra were recorded with a Perkin Elmer model Paragon 1000 PC spectrophotometer using the KBr disc method, at 4 cm^−1^ resolution and 100 scans between 4000 and 450 cm^−1^.

## 4. Conclusions

The use of coffee endocarp, sunflower, and vine shoots as precursors for activated carbon production is very motivating because the activated carbons produced have good porous and surface chemical properties, which are crucial to adsorb phenol and mercury species from aqueous solutions. Furthermore, the use of agricultural and industrial residues for the production of activated carbons has the advantage of minimizing the residues produced and the economical aspect of creating products of added value. The adsorption capacity observed compared quite well with other results published for similar systems. The maximum phenol uptake was 188 mg/g for pristine carbons (G838W) and 190 mg/g for TEDA-modified samples (G733WT). The maximum adsorption capacity for the adsorption of mercury species were 1104 mg/g for Hg^2+^ (V840ox), 771 mg/g for [HgCl_4_]^2−^ (Cf850T), and 966 mg/g for HgCl_2_ (Cf825T). The modification of the activated carbons with TEDA has had a significant positive impact on the adsorption of phenol and mercury species from aqueous solutions. Dispersive interactions among the impregnant delocalized electrons and both the phenolic ring and the mercury species maybe the cause. The adsorption of Hg^2+^ is incremented with the oxidisation of the activated carbons with nitric acid. It is also evident from the results obtained that we cannot make any generalisation about the adsorption mechanism; each case has to be analysed separately to take into account all relevant factors and their relative impact on the adsorption.

## Figures and Tables

**Figure 1 molecules-27-07345-f001:**
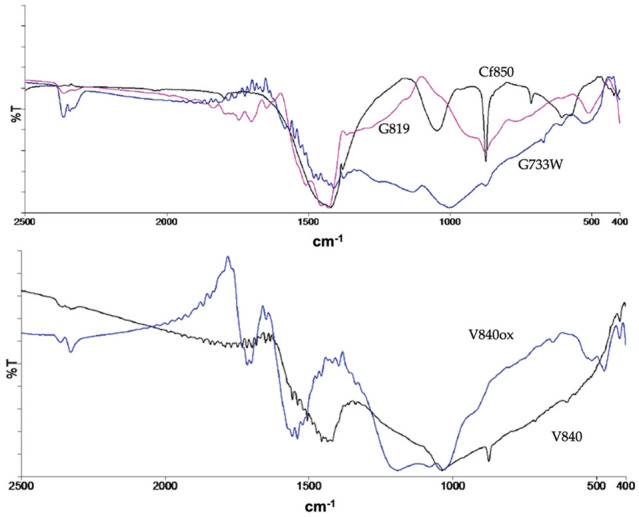
Representative FTIR spectra.

**Figure 2 molecules-27-07345-f002:**
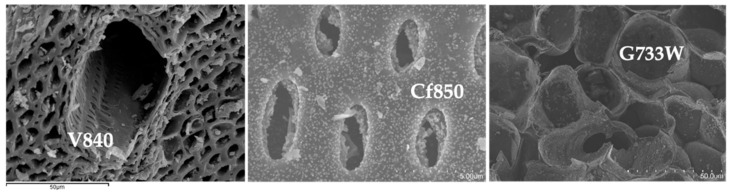
Representative SEM micrographs, samples V840, Cf850, and G733W.

**Figure 3 molecules-27-07345-f003:**
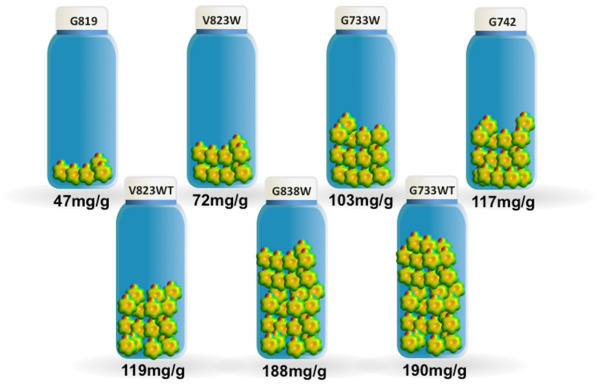
The maximum adsorption capacity for the phenol adsorption.

**Figure 4 molecules-27-07345-f004:**
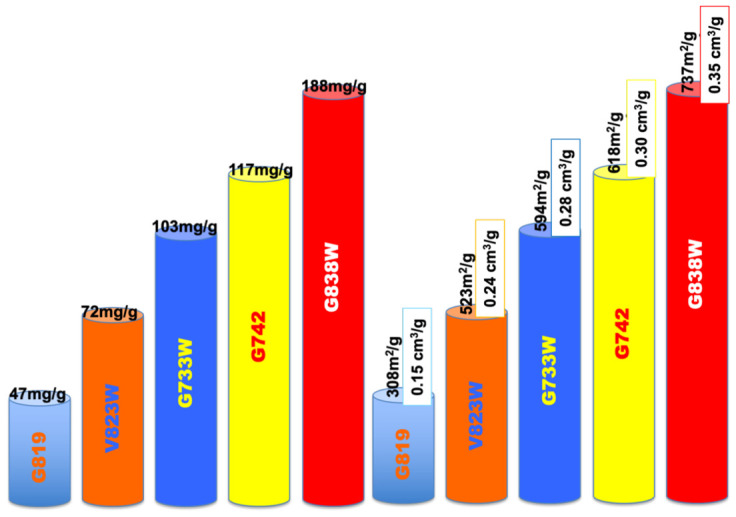
Impact of the porous structure on the phenol adsorption.

**Figure 5 molecules-27-07345-f005:**
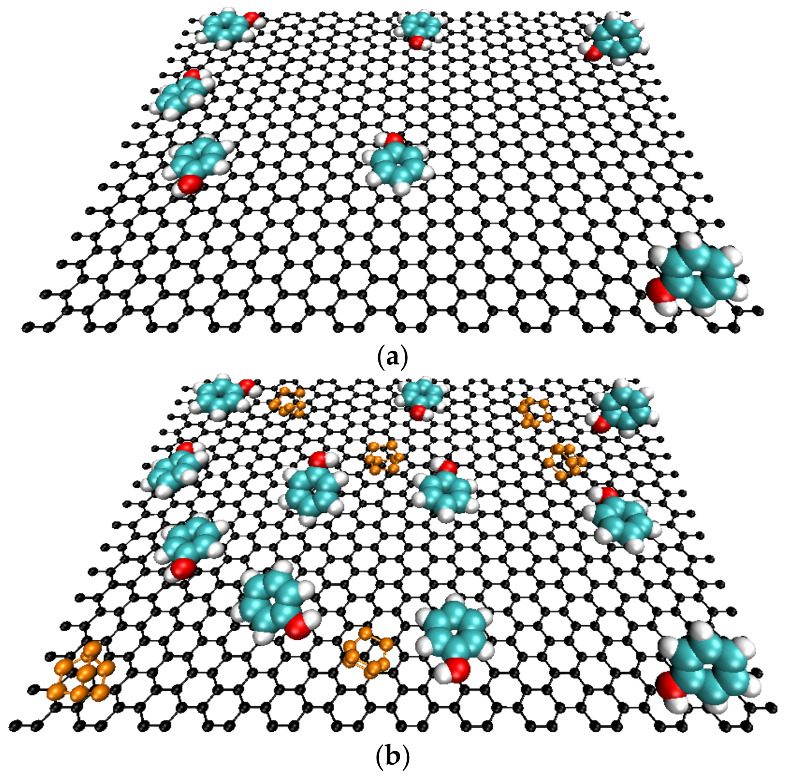
Schematic representation of the phenol adsorption at the activated carbon surface (**a**) pristine samples, (**b**) TEDA modified samples.

**Figure 6 molecules-27-07345-f006:**
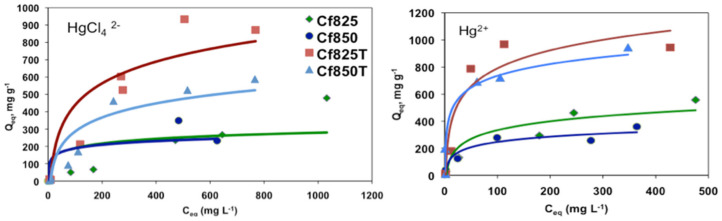
Representative adsorption isotherms for the adsorption of [HgCl_4_]^2−^ and Hg^2+^.

**Table 1 molecules-27-07345-t001:** Textural and chemical characterisations of the activated carbons.

Sample	Porosity				pzc
	BET	α_S_		DR	
	A_BET_ (m^2^g^−1^)	V_s_ (cm^3^g^−1^)	A_ext_ (m^2^g^−1^)	V_0_ (cm^3^g^−1^)	
Cf825	567	0.25	13	0.24	10.17
Cf825T	592	0.31	40	0.28	9.63
Cf850	796	0.36	19	0.35	10.07
Cf850T	626	0.30	48	0.29	9.65
V840 ^1^	956	0.44	26	0.40	9.71
V840ox ^1^	646	0.29	41	0.25	2.32
V823W	523	0.24	34	0.21	9.95
V823WT	358	0.19	31	0.16	9.60
G733W	594	0.28	26	0.25	9.86
G733WT	555	0.29	46	0.25	9.61
G838W	737	0.35	37	0.32	10.04
G742	618	0.30	71	0.27	9.52
G819	308	0.15	25	0.13	9.40

^1^ Mourão et al., 2001.

**Table 2 molecules-27-07345-t002:** Maximum adsorption capacity of mercury species in mg/g and net surface charge of the activated carbons.

Sample	Net Surface Charge	HgCl_2_	[HgCl_4_]^2−^	Hg^2+^
V840	+	584	204	896
V840ox	−	103	89	1104
Cf825	+	734	266	557
Cf825T	+	966	587	945
Cf850	+	204	232	277
Cf850T	+	290	771	968

## Data Availability

Not applicable.

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
