# Peer review of "Biomass Novel Adsorbents for Phenol and Mercury Removal"

_molecules, 2022, doi:10.3390/molecules27217345_

Round 1
Reviewer 1 Report
Dear authors, please find attached my comments to the manuscript

Author Response
We thank reviewer 1 for his/her constructive review that has help us improve our work; thank you very much. Here we answer the different points he/she has raised; as it can be seen, we have taken into account all them, and we have also reviewed other aspects to the manuscript. Two new references have been added and all the rest have been renumbered accordingly.
1- Some information should be added in the abstract, particularly referring to the reasons for the improvement observed after the modification of the activated carbons (increase of surface area and chemical interaction for phenol uptake, presence of electron-donor groups for mercury uptake)
Abstract: This paper reports the use of activated carbons made from novel agriculture and industrial wastes, namely sunflower, vine shoots and coffee endocarp, to remove two high priority contaminants: phenol and mercury species (under different forms) from aqueous solutions. The activated carbons were used as prepared and also modified with nitric acid and triethylenediamine in order to explore additional adsorption mechanisms. The results showed an interesting potential of the materials to be used for water decontamination as indicated by the mercury uptake up to 1104mg/g for Hg2+, 771mg/g for [HgCl4]2-, 966mg/g for HgCl2 and the maximum phenol adsorption capacity of 190mg/g. The modification with triethylenediamine led to a significant increase for the phenol and mercury adsorption reaching an increment of 85% for phenol and 250% for Hg2+; this was in both cases associated to the enhanced availability of electrodonating groups on the impregnated adsorbents
2- In the introduction the novelty of the work with respect to others in literature should be specified: is it in the substrate used for the preparation of the adsorbents? Is it in the overall characterization analysis performed? Please add in the new version
We understand that the novelty of the topis had to be better clarified. We have added the following paragraph to the last part of the introduction section.
In this context, this work aimed to evaluate the adsorption of phenol and mercury from dilute aqueous solutions onto ACs produced from agriculture and industrial wastes, namely vine shoots, sunflower and coffee endocarp and find the interactions governing the adsorption on each particular system. These three precursors meet the requirements of being abundant in the southwest of Iberian peninsula, and are biomass wastes that have been scarcely investigated to produce ACs. The thoughtful textural and chemical characterization of the adsorbents, and their subsequent impregnation with triethylenediamine, to further explore the enhancement of adsorption by the participation of additional surface groups, was related to removal capacities and optimal preparing conditions were identified.
3- The cited work in the introduction regarding phenol adsorption is quite old. Please add a more recent reference on the subject (see for example 10.1016/j.cattod.2020.08.013, where activated carbon is used to remove phenolic compounds and preserve catalyst stability)
We agree with the reviewers suggestion and we have included this reference in the new version of the manuscript; it really fits well since the cite defends microporosity as main responsible for phenol adsorption. We have added the reference at the discussion section, as follows:
According to the bibliography and on the authors own experience, one factor that clearly influences the adsorption performance is the pore volume and pore size distribution of the adsorbents. While the volume has to be enough to provide the adsorbate room to be adsorbed on several layers, if possible, the pore size distribution also has to be suitable to let the molecule access to the inner of the carbon matrix and impede blockage, in order to maximize removal efficiency. This effect has been clearly found from our runs, as it can be suggested from Figure 4, where both the phenol adsorption capacity and porosity parameters (BET apparent surface, m2/g and total pore volume, cm3/g) have been plotted as bar height, with different color for each run. In addition, in our case, all adsorbents are microporous, as deduced from the low contribution of external surface (AEXT, m2/g Table 1) what has been associated with favorable behaviour towards phenol compounds [21].
Row 310: please specify how the nitrogen stream was saturated with water
This part of the manuscript has been better clarified, and the following sentence has been added:
The nitrogen flow was saturated with water by passing the flow by a water reservoir that was heated to its boiling point with a flow rate of 0.2 g min-1.
2- I suggest adding the N2 ads/des isotherms to show the type I clearly: the manuscript is neither long nor full of figures, so I think it could be wise adding this information since it is cited
Unfortunately, the authors do not have access to the isotherms at this moment; the adsorption equipment is no longer available for the researchers, and the isotherms were saved there; the authors only conserve the porosity parameters taken from the adsorption unit software.
3- A comment on possible regeneration mechanisms could be added in the discussion of the results, since it is a key point for the utilization of such adsorbents
Although the regeneration of the adsorbents were not tested we believe that it can be accomplished by washing the adsorbents with warm water, wet oxygen regeneration pyrolysis or gasification; the authors have experience on these regeneration mechanisms applied to organic compounds. We have added a paragraph about this at the last part of the manuscript:
The adsorption results found here are promising and further studies might be devoted to explore the regeneration efficiency associated to each couple adsorbent-adsorbate; in this line, various regeneration methods including gasification, pyrolysis, wet oxygen regeneration or simple warn desorption could be explored.

Reviewer 2 Report
The article titled " Biomass novel adsorbents for Phenol and Mercury Removal" presents the development of a specific absorbent for the removal of Phenol and Mercury from wastewater.
I have the following recommendations.
You could write a little more about in which waters in real life these two elements, Phenol and Mercury, are found.
The chapter on materials and methods should come before the results since it is impossible to understand the content in that order. I suggest that the samples of adsorbents be described more precisely and given in a table where the meaning of the labels is explained, as it is difficult to follow what represents which sample.
Table 1 is difficult to understand, we suggest a description in Table of each porosity parameter.
The description under Figure 2 indicates a different pattern than indicated in the figure.
When describing the extraction of absorbents from natural materials such as sunflowers, vines, and coffee endocarp, it would be good to indicate the mass ratios between the input material and the initial amount of carbon.
What is missing from the discussion is a comparison of adsorption capacities between commercial activated carbon and that used in your research. It would also be good to point out the realistic possibilities of implementing this type of adsorption process in terms of economic feasibility.
The article can be published with minor corrections
Author Response
Answers to reviewer 2.
We thank reviewer 2 for his/her constructive review that has help us improve our work; thank you very much. Here we answer the different points he/she has raised; as it can be seen, we have taken into account all them, and we have also reviewed other aspects to the manuscript. Two new references have been added and all the rest have been renumbered accordingly.
- When describing the extraction of absorbents from natural materials such as sunflowers, vines, and coffee endocarp, it would be good to indicate the mass ratios between the input material and the initial amount of carbon.
The overall solid yield of the process could be calculated can be calculated from the burn-off degree, knowing that the yield of the carbonization is around 50%. We have included an explanatory sentence adding this information in section 3.1.
The BO was determined as the relation between the mass consumed in reference to the initial char mass; char solid yield for the carbonization conditions set here, was around 50%.
- What is missing from the discussion is a comparison of adsorption capacities between commercial activated carbon and that used in your research. It would also be good to point out the realistic possibilities of implementing this type of adsorption process in terms of economic feasibility.
We have included in our work a comparison between our adsorption results and those shown by other authors, for phenol (refs. 18, 19 and 20) or some of us, in previous works with other researchers (refs. 21, 23 and 25). Regarding the economic feasibility study, the authors are not sure if the reviewer means an economic study on the AC production process or on the of the adsorption process. We really agree this is really important to assess the feasibility of the process industrial implementation, and we are planning to investigate on this part in the future, but the current work is devoted to understand the mechanisms involved on the adsorption of these pollutants; we think that the feasibility study devoted a profound analysis that is out of the scope of our manuscript.
- In the introduction the novelty of the work with respect to others in literature should be specified: is it in the substrate used for the preparation of the adsorbents? Is it in the overall characterization analysis performed? Please add in the new version
From the authors point of view, one interesting point of this work is related to the precursors used. The three of them are very typical and abundant in south west Spain and central Portugal, the areas where the authors live. The three of them mean lucrative business at the moment and at the same time almost null research has been performed regarding AC production with them.
The research for methods to reduce the phenol and mercury discharges and for its removal from waters streams are still needed. In that sense, the innovative aspect of the work now reported is the modification of activated carbons with triethylenediamine to remove phenol and mercury from aqueous solutions; moreover, styding the adsorption behaviour of the three Hg species is a highlighting novelty too. The results are of quality and showed that the proposed method is an added value to this field of knowledge.
We have tried to better emphasize this aspect in the new version of the manuscritpt, by adding several words or sentences in different sections of the manuscript, such as:
In this context, this work aimed to evaluate the adsorption of phenol and mercury from dilute aqueous solutions onto ACs produced from agriculture and industrial wastes, namely vine shoots, sunflower and coffee endocarp and find the interactions governing the adsorption on each particular system. These three precursors meet the requirements of being abundant in the southwest of Iberian peninsula, and are biomass wastes that have been scarcely investigated to produce ACs. The thoughtful textural and chemical characterization of the adsorbents, and their subsequent impregnation with triethylenediamine, to further explore the enhancement of adsorption by the participation of additional surface groups, was related to removal capacities and optimal preparing conditions were identified.
Or
The study of three different mercury species, namely HgCl2, Hg2+ and [HgCl4]2- , is relevant to mimic what happens in the water streams where mercury usually exists in the form of complexed species, which can be negative, neutral or positive, and is not usually tackled in the bibliography.
